# A compact co-aperture dual-sense circularly polarized antenna for simultaneous transmit and receive systems

**Tan Dao-Duc[1], Duc-Nguyen Tran-Viet ⊙[2], Dat Nguyen Tien[1], Dinh Nguyen Quoc[3], Tung The-Lam Nguyen[4], Hung Tran-Huy ⊙[1] ***

**1** Faculty of Electrical and Electronic Engineering, PHENIKAA University, Hanoi, Vietnam, **2** Faculty of Radio-Electronic Engineering, Le Quy Don Technical University, Hanoi, Vietnam, **3** Vietnam-Japan International Cooperation Center for Science and Technology (VJIC), Le Quy Don Technical University, Hanoi, Vietnam, **4** IT Department, Greenwich Vietnam, FPT University, Hanoi, Vietnam

* hung.tranhuy@phenikaa-uni.edu.vn

**Data Availability Statement:** All relevant data are within the manuscript.

## Abstract

This paper proposes a compact design of dual-sense circularly polarized (CP) antenna for simultaneous transmit (Tx) and receive (Rx) communication systems. The primary radiating aperture of the proposed antenna is a 2 × 2 unit-cell metasurface (MS). The MS is excited by the asymmetric patch in the center, which acts as the CP source of the whole antenna structure. By properly tuning the feeding positions, dual-sense CP with high isolation can be achieved. For verification, an antenna prototype with compact dimensions of 0.36λ × 0.36λ × 0.02λ (λ is the free-space wavelength at the center operating frequency) is fabricated and measured. The measured operating bandwidth is 1.6% (2.45–2.49 GHz), in which the reflection and transmission coefficients are less than—10 dB and the axial ratio is lower than 3 dB. Within this band, the maximum isolation value is 39 dB, and the peak gain is 5.7 dBi.

## Introduction

In modern wireless communications, simultaneous transmit and receive (STAR) or in-band full-duplex (IBFD) technology has attracted lots of attention since it allows the system to transmit and receive at the same frequency and same time [1]. Various types of STAR antennas have been presented in the literature using monopole [2], dipole [3], and microstrip patch structures [4]. For different applications, the requirements for radiation pattern, gain, as well as antenna profile are different. Along with this, antenna polarization is also another critical issue. Compared to the linearly polarized (LP) antenna, the circularly polarized (CP) antenna is preferred due to its unique features for both multipath and line-of-sight propagations. This paper only focuses on STAR antennas with dual-sense CP radiation using microstrip patch structures to achieve low-profile and directional radiation beams.

Depending on the primary radiating aperture, STAR antennas can be divided into two different types, including multi-aperture and co-aperture antennas. Firstly, the multi-aperture

**Funding:** The author(s) received no specific funding for this work.

**Competing interests:** The authors have declared that no competing interests exist.

case consists of the Tx and Rx antennas that have different radiating apertures and are separated from each other. To improve the Tx-Rx isolation, various techniques, including self-decoupling [5], defective ground structure [6], parasitic elements [7], and MS [8, 9], have been proposed. Nonetheless, using multi-aperture for STAR antenna suffers from a critical drawback of largely occupied space, which is not suitable for space-constrained systems such as short-range motion sensing and gesture sensing.

To overcome the deficiency of the first type, the Tx and Rx segments use the same radiating aperture known as the co-aperture STAR antenna. However, suppressing mutual coupling for this type of antenna is more challenging than the multi-aperture one due to the co-aperture property and closely located ports. Dual CP can be easily achieved with the aid of a branch line coupler [10–13]. Such designs employ the coupler as the feeding structure, and the antenna can provide either right-hand CP (RHCP) or left-hand CP (LHCP), depending on the excitation port. Although wideband performance can be achieved, their critical disadvantages are large overall sizes and complicated structures with multiple printed layers. Smaller sizes and less complicated designs could be obtained with different feeding techniques. Dual CP are realized when the patch antennas are fed by two L-strips [14], coplanar waveguide transmission line [15], multiple slots [16, 17], a suspended strip line [18], or U-shaped slots [19, 20]. Alternatively, techniques using the MS [21, 22] or a substrate-integrated waveguide [23, 24] have also been demonstrated as promising candidates for low-profile, wideband, dual-CP antenna. However, the most challenging aspect is still large overall dimensions, which make them less appealing for compact electronic devices.

In this paper, the dual CP antenna with size miniaturization is concentrated. The proposed STAR antenna employs a $2 \times 2$ unit-cell MS as the co-aperture to achieve compact size. Further size reduction is obtained by embedding a squared-ring slot into the conventional squared-shaped unit cell. To achieve dual CP radiation, the MS is fed by the asymmetric patch in the center, and by properly adjusting the feeding positions, high port isolation is obtained. In comparison with the related works, the proposed antenna offers a small size while exhibiting good Tx-Rx isolation and consistent Tx and Rx radiation patterns.

## Single-port MS-based antenna

Fig 1 shows the configuration in terms of the top and side views of the single-port MS-based antenna. The antenna with three layers of ground, CP source, and MS is designed on two Taconic RF-35 substrates, in which each substrate has a dielectric constant of 3.5 and thickness of 1.52 mm. It is noted that the operating principle and the key optimizing parameters of this antenna have been thoroughly investigated and presented in [25], which is why they are not mentioned in this paper. In general, this design has the advantages of compact size while possessing high gain radiation. Fig 2 shows the performance in terms of reflection coefficient, axial ratio (AR), and broadside gain of this antenna. As observed around 2.45 GHz, the antenna shows good impedance matching with $|S_{11}|$ of less than –10 dB and good CP radiation with AR of lower than 3 dB. Meanwhile, the broadside gain is about 7 dBi.

## Miniaturization of single-port MS-based antenna

It has been demonstrated in [25] that the MS is a primary radiating aperture of the MS-based antenna, and the patch acts as the CP source. To minimize the antenna size, the MS layer should be redesigned. Fig 3 shows the geometry of the conventional MS (as in the previous Section) and the proposed MS. For the proposed one, square-shaped slots are embedded into the unit cells. Overall, the dimensions of the proposed MS are smaller than the conventional one while having similar operating frequencies.

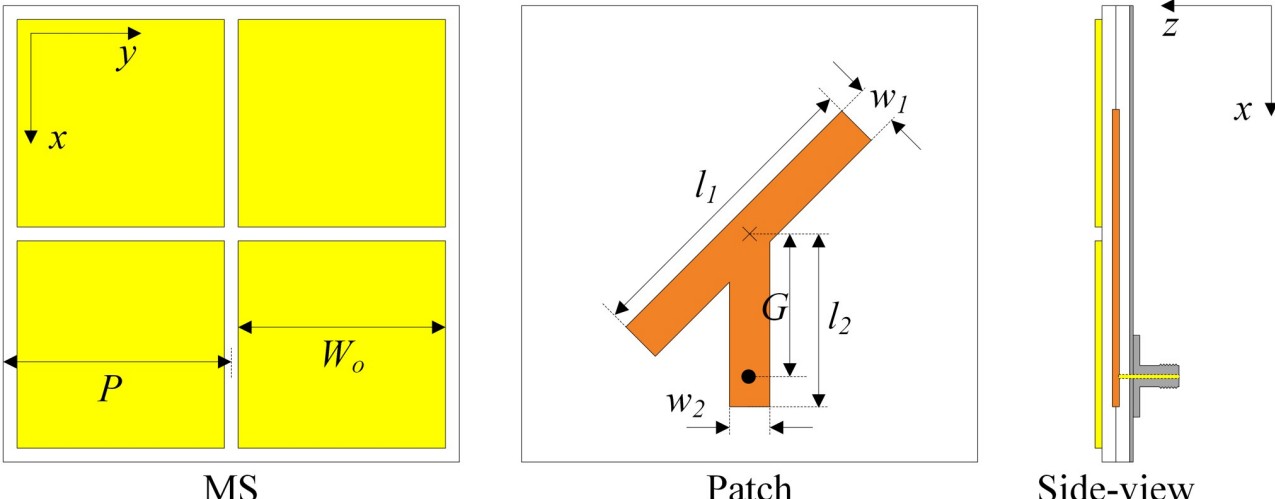

**Fig 1. Geometry of the proposed single-port MS-based CP antenna.** $P = 27.5$, $W_0 = 27.1$, $h_1 = h_2 = 1.52$, $l_1 = 37$, $w_1 = 5$, $l_2 = 21$, $w_2 = 5$, $G = 20$ (unit: mm).

For demonstration, the characteristic mode analysis (CMA) on the conventional and proposed MSs is carried out using the MoM-based CMA tool in the commercial simulation software CST MWS [26, 27]. Note that the MSs can produce good CP radiation at an arbitrary frequency when the modal significance at that frequency is equal to 1, and two orthogonal modes with similar broadside directivity are required. The modal significances of the conventional and proposed MSs are illustrated in Fig 4. Here, the first 4 modes from 2.2 to 3.2 GHz are calculated and sorted at 2.4 GHz. As seen in the interested frequency range around 2.5 GHz, both MSs have similar modal significance values of approximately 1 for Mode 1 and Mode 2. Besides, the current distributions and radiation patterns demonstrate that these modes are orthogonal, and their broadside directivity values are almost similar. These results prove that the proposed MS has the potential to work effectively around 2.5 GHz, which is

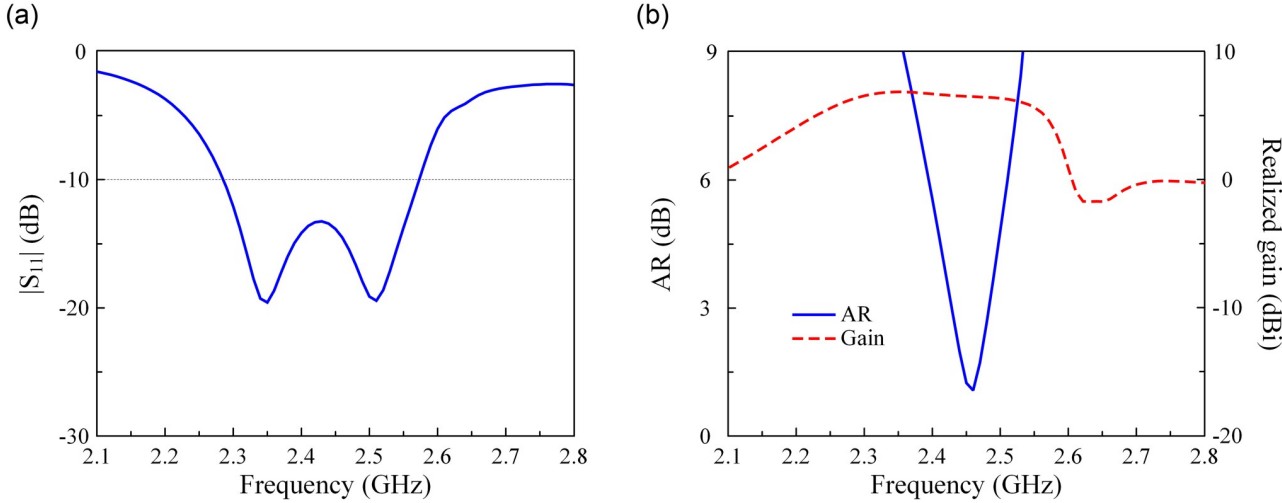

**Fig 2. Simulated performance of different antennas.** (a) $|S_{11}|$, (b) AR and realized gain.

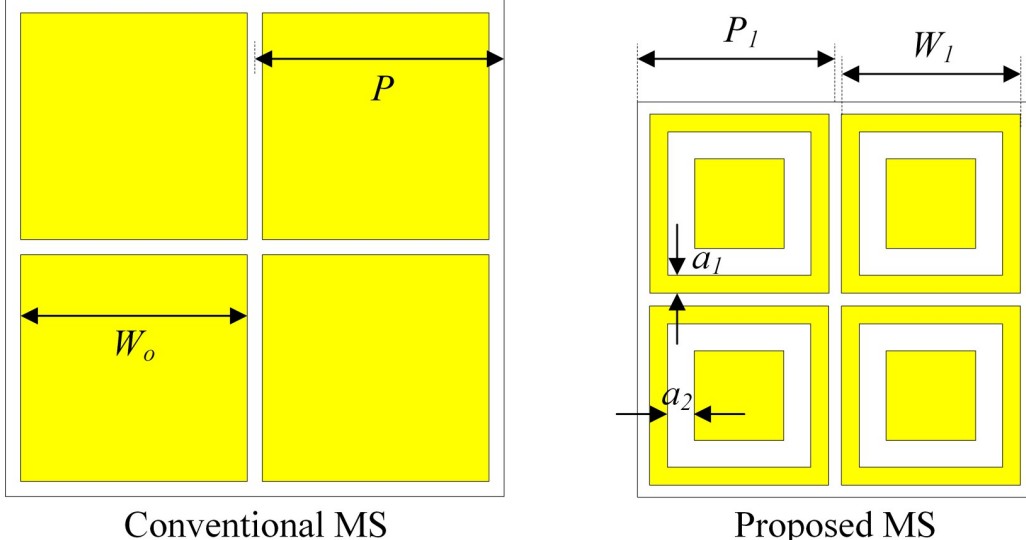

**Fig 3. Geometry of the conventional and proposed MSs.** The dimensions of the conventional and proposed MSs are $P = 27.5$, $W_0 = 27.1$, $P_1 = 22$, $W_1 = 20$, $a_1 = 1.7$, $a_2 = 2.5$ (unit: mm).

similar to the conventional MS. However, the proposed MS has a smaller size in comparison with the conventional MS, which are 22 mm and 27 mm, respectively.

To demonstrate the effectiveness of the proposed MS, Fig 5 shows the geometry of the proposed single-port MS-based antenna with size miniaturization. Similar to the design in the previous Section, the antenna in this Section is also designed on two 1.52-mm-thick Taconic substrates for a fair comparison. The performance of the proposed MS-based antenna with miniaturized sizes is depicted in Fig 6. The simulated data indicates that the proposed antenna has good performance around 2.45 GHz.

## Dual-sense CP two-port MS-based antenna

In this Section, a CP antenna with the capability of producing RHCP and LHCP radiations is proposed. This antenna is developed from the design shown in the previous Section. Fig 7 shows the geometry of the proposed dual-sense CP antenna. The antenna is fed by two different ports designated as Port-1 and Port-2 through Stub-1 and Stub-2. When Port-1 is excited, the antenna has LHCP radiation. In contrast, RHCP is dominant when Port-2 is excited. The performance of the proposed dual-sense CP is shown in Fig 8. As observed, the antenna well operates at 2.46 GHz with reflection and transmission coefficients better than -32 dB. Meanwhile, the AR value is less than 3 dB, and the gain is about 6.0 dBi. Besides, the Envelop Correlation Coefficient (ECC) is also evaluated. Here, around 2.46 GHz, the ECC value is about 0.002, which is significantly lower than the acceptable value of 0.5.

Next, the decoupling mechanism of the proposed antenna is discussed. The MS plays an important role in determining the inter-port isolation of the proposed design. Fig 9 shows the current distributions on the CP source at 2.46 GHz for different cases of with and without MS. It can be seen obviously that with the presence of the MS, the field on the CP source is significantly redistributed and the intensity field is weak at the Port-2 feeding branch. This is due to the MS is designed to well resonate at 2.46 GHz as observed in Fig 4b with high modal significance of approximately 1. At this frequency, the electromagnetic field from the CP source

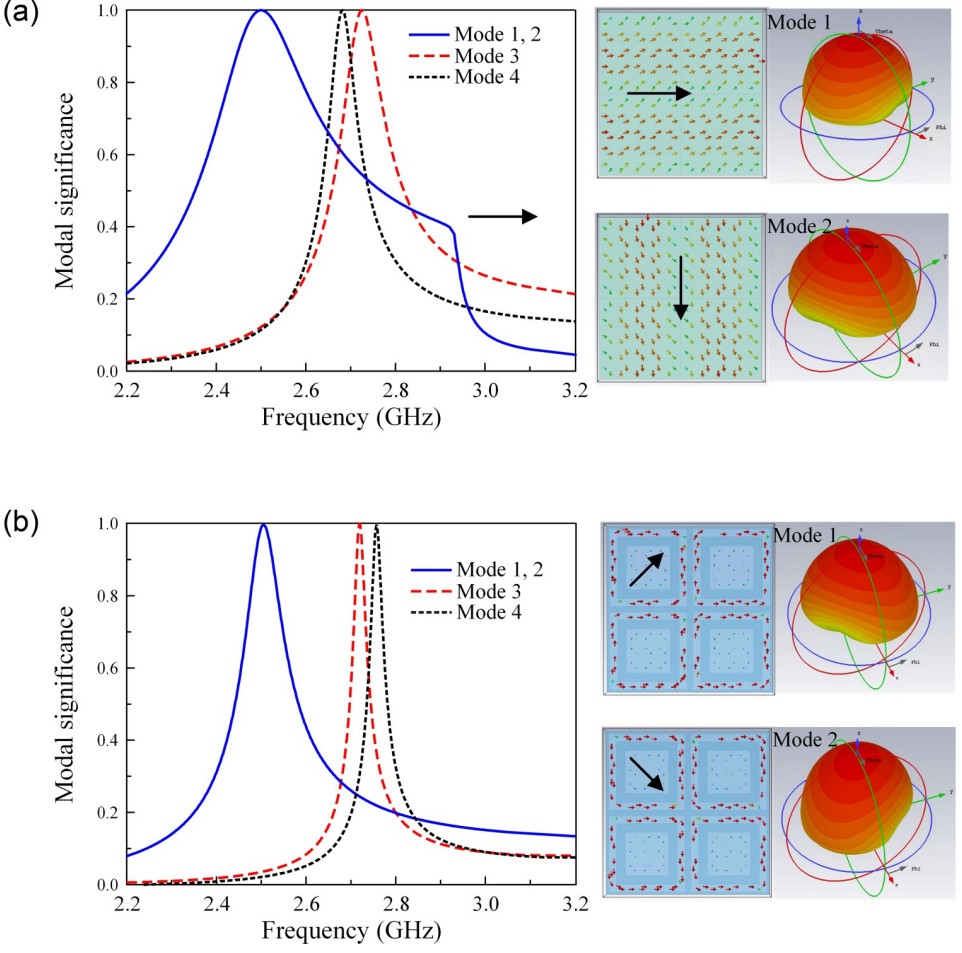

**Fig 4. Modal significance of (a) conventional MS and (b) proposed MS.**

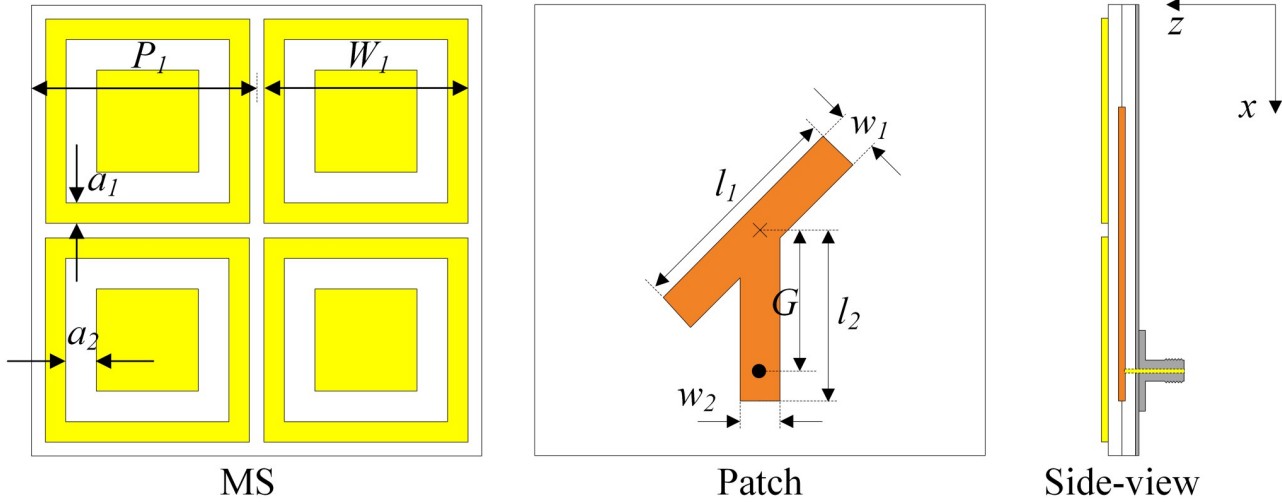

**Fig 5. Geometry of the proposed single-port MS-based CP antenna.** $P_1$ = 22, $W_1$ = 20, $a_1$ = 1.7, $a_2$ = 2.5, $l_1$ = 13.2, $w_1$ = 3.8, $l_2$ = 13.4, $w_2$ = 4, $G$ = 12.4 (unit: mm).

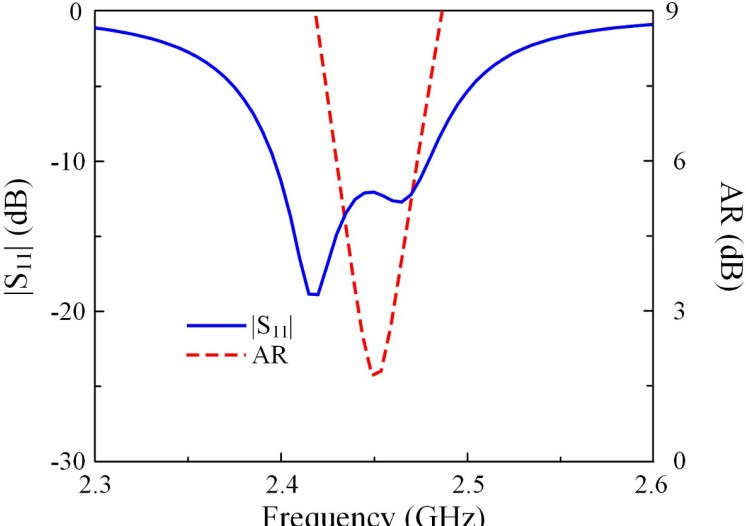

**Fig 6. Simulated performance of the proposed MS-based antenna with size miniaturization.**

tends to couple with the MS, rather than the Port-2 feeding branch, leading to high inter-port isolation. Out of the frequency band around 2.46 GHz, the field distribution on the CP source alters significantly. Fig 10 shows the field distributions on the CP source at different frequencies. At 2.4 and 2.5 GHz, strong currents are observed on the Port-2 feeding branch, resulting in high isolation. This is attributed to the MS, which does not work at these frequencies due to the low modal significance values of less than 0.4 (shown in Fig 4b). Thus, the isolation performance at these frequencies is worse than at 2.46 GHz.

Since the current distribution in the Port-2 feeding branch is quite weak around 2.46 GHz, the feeding position of Port-2 is tuned to achieve the best isolation performance. Fig 11 shows

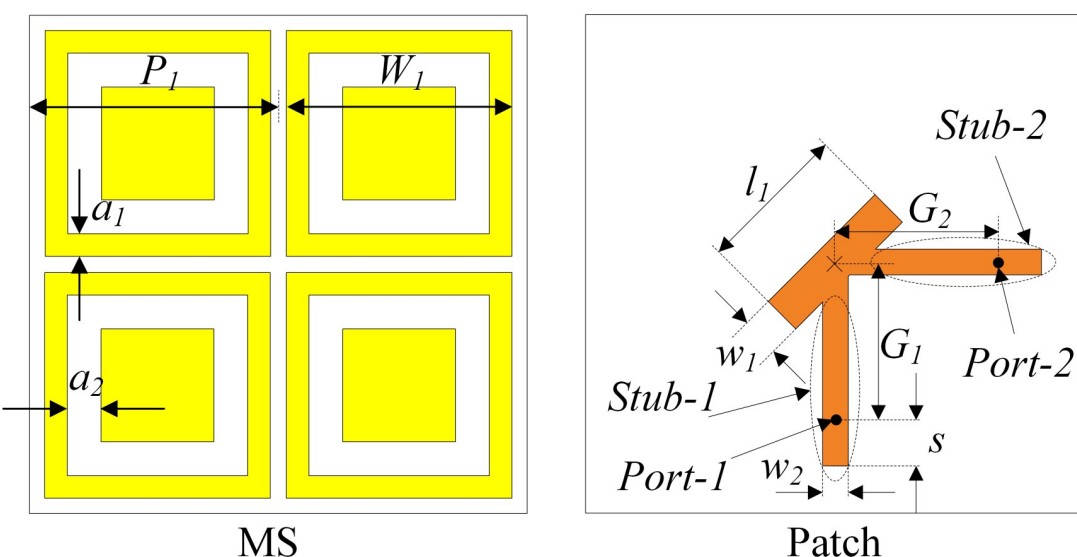

**Fig 7. Geometry of the proposed single-port MS-based CP antenna.** $P_1 = 22$, $W_1 = 20.3$, $a_1 = 1.9$, $a_2 = 2.5$, $l_1 = 12$, $w_1 = 3$, $s = 6$, $w_2 = 1.2$, $G_1 = G_2 = 12$ (unit: mm).

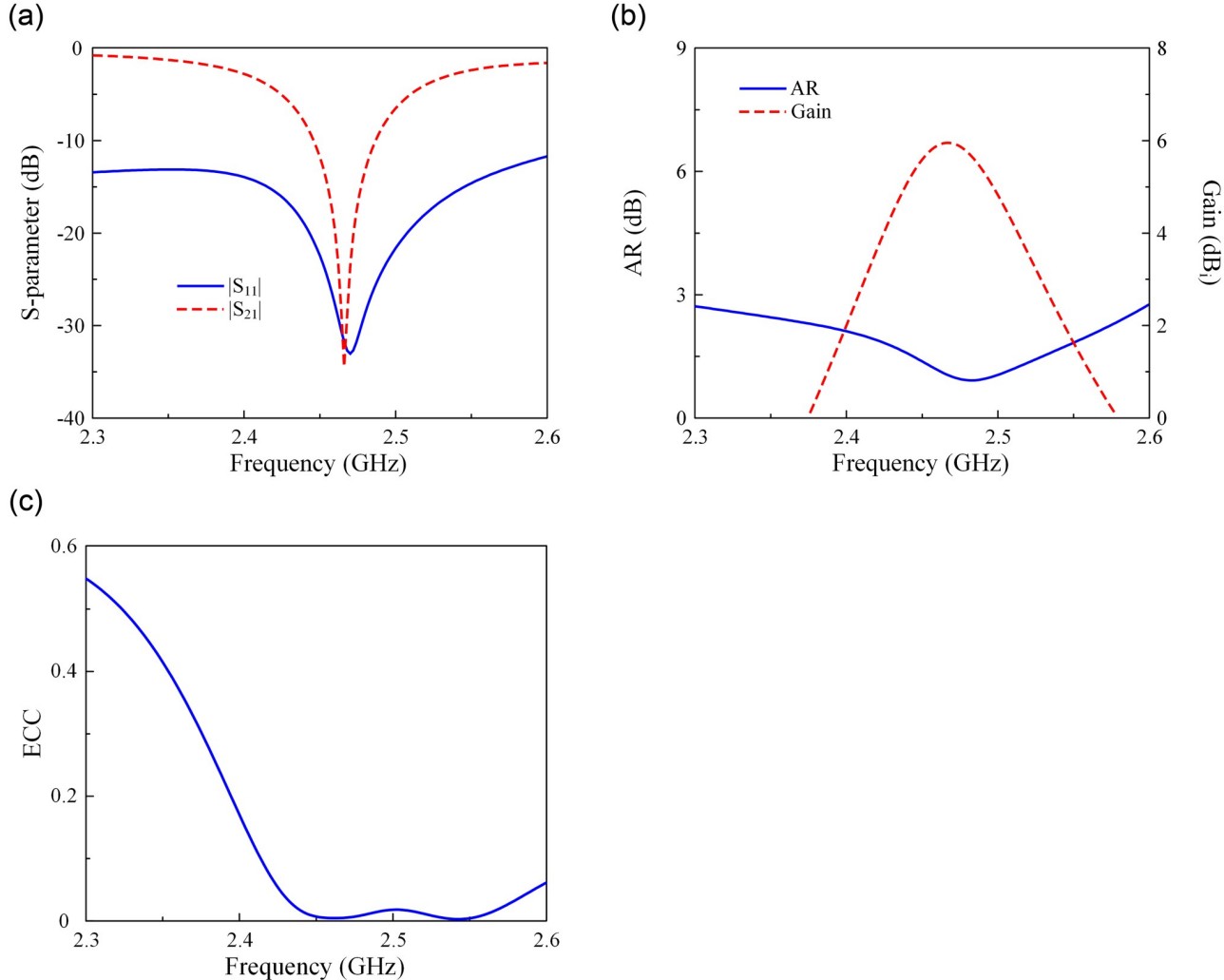

**Fig 8. Simulated (a) S-parameter, (b) AR and gain, (c) ECC of the proposed antenna.**

the simulated S-parameter of the proposed antenna with different values of $G_2$. The data indicates that a smaller value of $G_2$ results in lower isolation between two ports. With respect to $|S_{11}|$, changing $G_2$ has an insignificant effect on the matching around 2.46 GHz. It means that the effect of Stub-2 on the matching performance with Port-1 excitation is minor.

Next, the impedance matching of the proposed antenna is considered. It has been found that the impedance matching can be conveniently adjusted by tuning the feeding position *s*. Fig 12 presents the simulated S-parameter against the variations of *s*. Theoretically, changing *s* will change the capacitance and inductance of the antenna. Thus, the impedance matching can be adjusted. According to the results in Fig 12, the matching feature is significantly affected by *s*, which improves when *s* increases from 2 to 6 mm. Out of this range, the matching is degraded. In terms of isolation, it is almost stable with different values of *s*.

## Measurement results

To validate the proposed design concept, an antenna prototype was fabricated and measured. The photographs of the fabricated prototype with different layers are presented in Fig 13. The

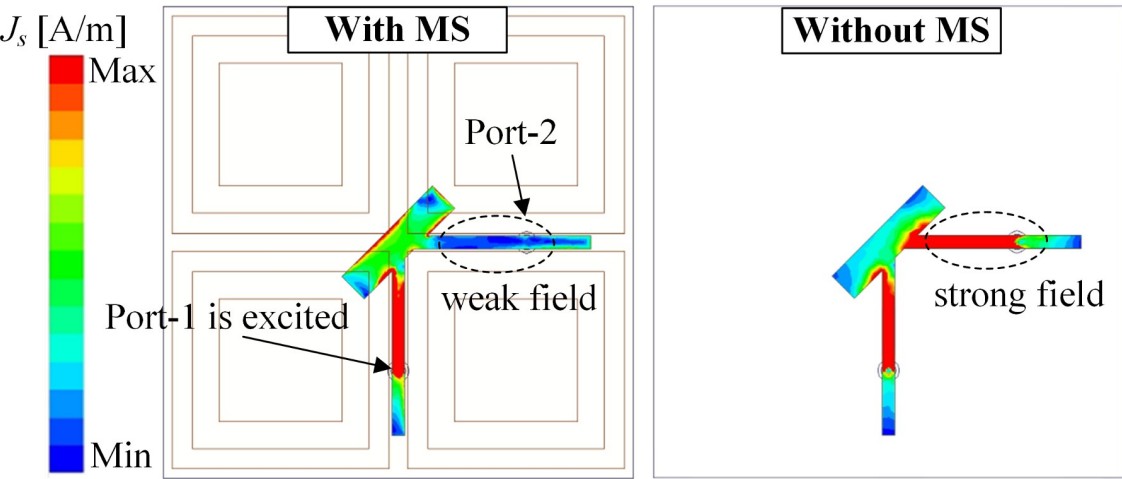

**Fig 9. Simulated current distributions on the patch at 2.46 GHz with and without MS.**

S-parameter measurement is implemented with a network analyzer. At the same time, the far-field performance is characterized in an anechoic chamber. In general, there is a similarity between the simulated and measured results. The small discrepancy could be attributed to the tolerance of fabrication and measurement setup.

The simulated and measured S-parameter results in terms of reflection coefficient and transmission coefficient are shown in Fig 14. Meanwhile, the AR and broadside gain values are illustrated in Fig 15. The measured data indicates that the proposed antenna shows good performance around 2.46 GHz. The measured operating bandwidth (BW) is from 2.45 to 2.49 GHz, equivalent to about 1.6%, in which the reflection and transmission coefficients are less than –10 dB, and the AR is smaller than 3 dB. Meanwhile, the maximum isolation value is 39 dB, and the peak broadside gain is about 5.7 dBi. Regarding the antenna efficiency, the simulated value is about 84%. In measurement, the antenna efficiency was not measured due to the limit of the chamber. However, it can be estimated to about 75% based on the realized gain measurements.

The simulated and measured radiation patterns at 2.46 GHz in two principal planes of $x - z$ and $y - z$ are plotted in Fig 16. The radiation patterns are LHCP and RHCP with Port-1 and

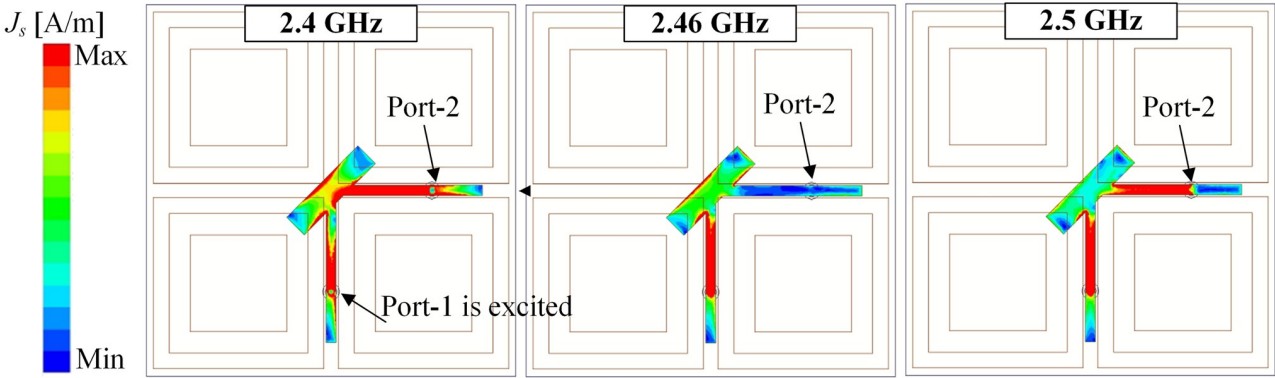

**Fig 10. Simulated current distributions on the patch at different frequencies.**

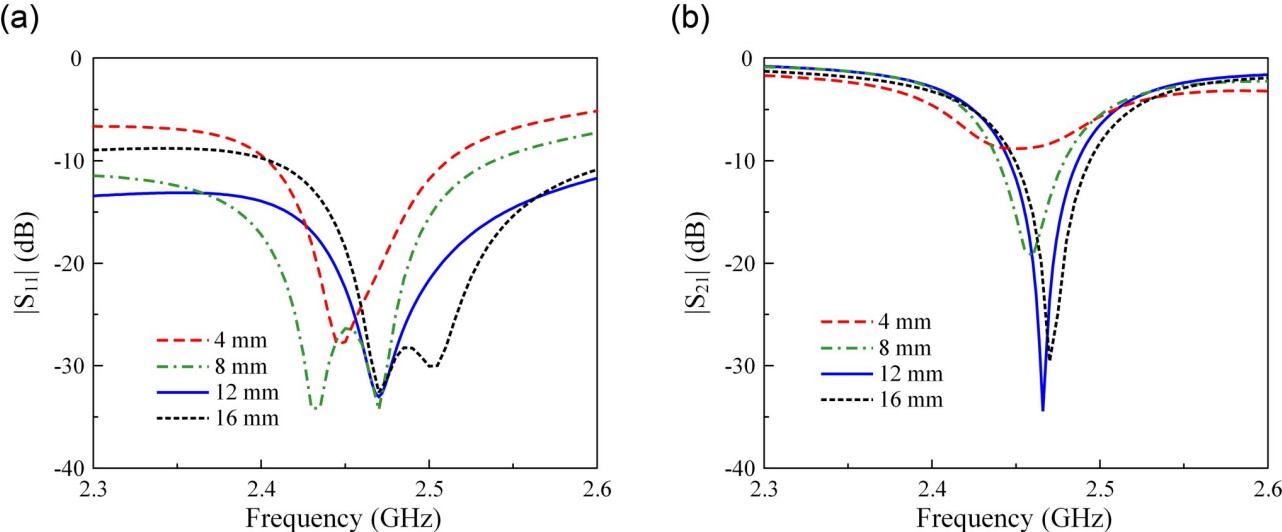

**Fig 11. Simulated (a) $|S_{11}|$ and (b) $|S_{21}|$ of the proposed antenna for different values of $G_2$.**

Port-2 excitations, respectively. Due to the symmetrical geometry, only the results for Port-1 excitation are presented. In the forward direction, the polarization isolation defined by the difference between the co-polarization and cross-polarization is about 22 dB. The front-to-back ratio is about 14 dB.

To demonstrate the advantages of the proposed antenna, Table 1 illustrates a performance comparison among the dual-sense CP antennas. Note that the operating BW is specified by a 10-dB isolation level, and it also covers the 3-dB AR BW and -10 dB impedance BW. The strong features of the proposed antenna are compact size and high isolation while possessing a comparable gain. It is worth noting that designs [16, 17, 19, 21, 22] use slotted fed as a feeding

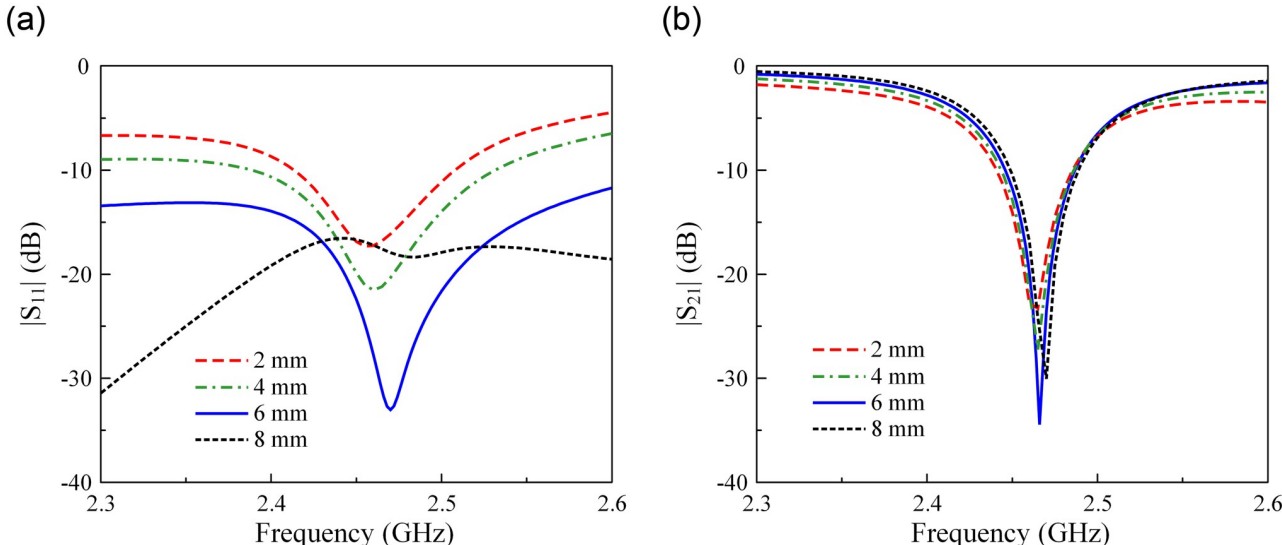

**Fig 12. Simulated (a) $|S_{11}|$ and (b) $|S_{21}|$ of the proposed antenna for different values of $s$.**

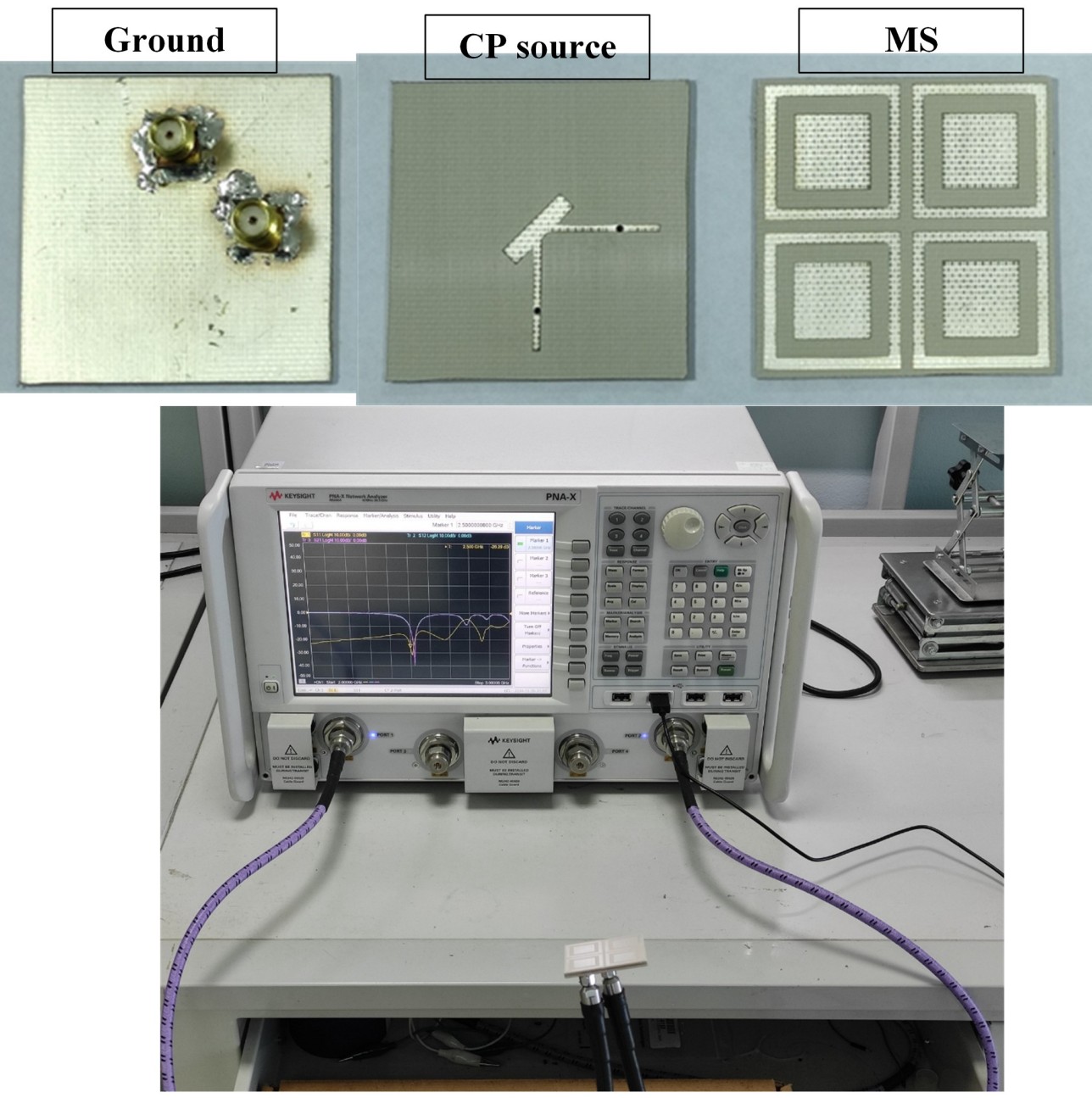

**Fig 13. Photographs of the fabricated antenna prototype.**

method, which significantly reduces the integration of the antenna into multi-layer electronic circuits. This is due to the required distance from the antenna to other circuit layers to avoid interference in the antenna's performance. A similar drawback is also observed in [20] with a double U-slot embedded into the ground. It is also noted that the proposed antenna has narrowband operation due to the high Q-factor. As this paper focuses on compact designs, there is a trade-off with operating BW. However, despite having the smallest overall dimensions, the proposed antenna can achieve high isolation and comparable gain as well.

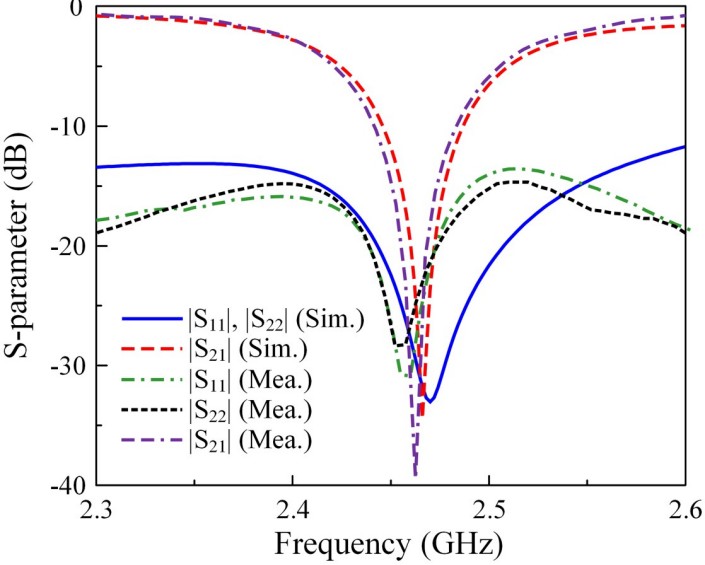

**Fig 14. Simulated and measured S-parameter of the proposed antenna.**

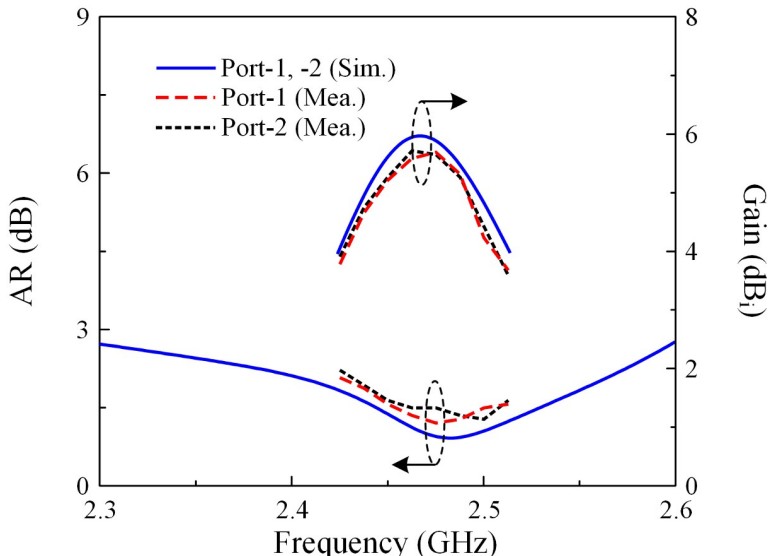

**Fig 15. Simulated and measured AR and gain of the proposed antenna.**

## Conclusion

A compact design of dual-sense CP antenna for simultaneous transmit and receive communication systems is presented and investigated in this paper. The final design has a compact overall size of 0.36λ × 0.36λ × 0.02λ (λ is the free-space wavelength at the center operating frequency). Measurements have been implemented on the fabricated antenna prototype to demonstrate the feasibility of the design concept. The measured results show an operating bandwidth of 1.6% (2.45–2.49 GHz). Across this operating band, the maximum isolation value

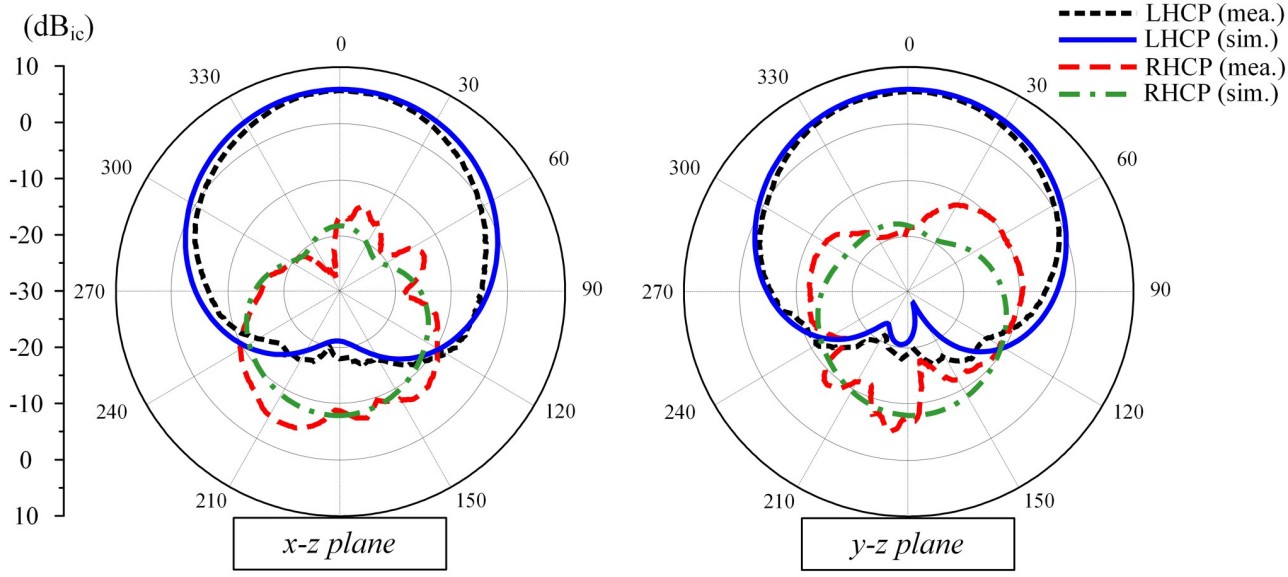

**Fig 16. Simulated and measured radiation patterns at 2.46 GHz.**

**Table 1. Performance comparison among co-aperture dual-sense CP antennas.**

| Ref | Overall size ($\lambda_c$) | Radiating aperture | Feeding structure | BW (%) | Maximum isolation (dB) | Peak gain (dBi) |
|---|---|---|---|---|---|---|
| [16] | $0.96 \times 0.96 \times 0.20$ | Patch | Slotted fed | 19 | - | 9.6 |
| [17] | $0.39 \times 0.39 \times 0.04$ | Patch | Slotted fed | 11 | 33 | 4.9 |
| [18] | $0.83 \times 0.83 \times 0.12$ | Patch | Suspended strip line | 15.7 | 22 | 7.8 |
| [19] | $0.72 \times 0.72 \times 0.09$ | Patch | Slotted fed | 18.5 | 35 | 7.9 |
| [20] | $0.43 \times 0.43 \times 0.02$ | Patch | Directly fed + slot | 2.5 | 34 | 5.4 |
| [21] | $0.73 \times 0.73 \times 0.07$ | MS | Slotted fed | 12.4 | 21 | 7.1 |
| [22] | $0.85 \times 0.85 \times 0.05$ | MS | Slotted fed | 31.3 | 25 | 7 |
| Prop. | $0.36 \times 0.36 \times 0.02$ | MS | Directly fed | 1.6 | 39 | 5.7 |

of 39 dB is achieved, and the peak broadside gain is 5.7 dBi. Additionally, the proposed antenna has 10-MHz BW with isolation of better than 20 dB. This could be suitable for STAR systems operating with Zigbee and Bluetooth standards.

## Author Contributions

**Investigation:** Tan Dao-Duc, Duc-Nguyen Tran-Viet, Dat Nguyen Tien.

**Methodology:** Tan Dao-Duc, Tung The-Lam Nguyen.

**Supervision:** Dinh Nguyen Quoc, Tung The-Lam Nguyen.

**Validation:** Tan Dao-Duc, Dinh Nguyen Quoc.

**Writing – original draft:** Tan Dao-Duc, Duc-Nguyen Tran-Viet, Dat Nguyen Tien.

**Writing – review & editing:** Hung Tran-Huy.

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
