## [Decision Letter · Decision Letter 0]

26 Mar 2024

PONE-D-24-05575A compact co-aperture dual-sense circularly polarized antenna for simultaneous transmit and receive systemsPLOS ONE

Dear Dr. Tran-Huy,

Thank you for submitting your manuscript to PLOS ONE. After careful consideration, we feel that it has merit but does not fully meet PLOS ONE’s publication criteria as it currently stands. Therefore, we invite you to submit a revised version of the manuscript that addresses the points raised during the review process.

We look forward to receiving your revised manuscript.

Kind regards,

Maharana Pratap Singh, Ph.D.

Academic Editor

PLOS ONE

Journal Requirements:

2. Please note that PLOS ONE has specific guidelines on code sharing for submissions in which author-generated code underpins the findings in the manuscript. In these cases, all author-generated code must be made available without restrictions upon publication of the work. 

Please review our guidelines at https://journals.plos.org/plosone/s/materials-and-software-sharing#loc-sharing-code and ensure that your code is shared in a way that follows best practice and facilitates reproducibility and reuse.

**Additional Editor Comments:**

Dear Dr. Hung Tran-Huy,

I am writing in regards to manuscript No PONE-D-24-05575 entitled “A compact co-aperture dual-sense circularly polarized antenna for simultaneous transmit and receive systems” which you submitted to the PLOS ONE.

Your manuscript has been reviewed by several experts in the field.

Based on the recommendations given by reviewers found at the bottom of this letter, I suggest you to please address the questions raised by the reviewers.

Reviewer 1:

This paper proposes a compact design of dual-sense circularly polarized for simultaneous transmit (Tx) and receive (Rx) communication systems. The high Tx-Rx isolation can be achieved by tuning the feeding positions. There are some major issues that raise my concerns.

1. Undoubtedly, the isolation of the transmitting and receiving ports is a critical parameter for a STAR antenna. Although the article explains how to achieve high Tx-Rx isolation by merely adjusting the feed position, its decoupling mechanism is not explicitly described. To enhance the reader's comprehension, kindly provide a comprehensive description of this. Also, could you please state what challenges there are for the decoupling problem of dual circularly polarized antennas with the same aperture?

2. In contrast to the co-aperture dual-sense CP antennas that are comparable in size, the antenna proposed in the article possesses the most limited bandwidth. Could you please provide an explanation as to why?

3. More comprehensive results need to be provided so that readers can clearly understand the effectiveness of the proposed antenna, such as antenna efficiency, envelope correlation coefficient, and radiation pattern of both ports.

4. In Figure 12, it can be clearly observed that the tested S21 is better than the simulated one. Is this phenomenon coincidental? If not, then elucidate this.

5. There are many grammar mistakes in this paper. For example, The last paragraph of the introduction. "In this paper, the dual CP antenna with size miniaturization is concentrated. The proposed STR antenna employs 2 ×2 unit-cell MS as the co-aperture to achieve compact size. ". Page 4, third line of the first paragraph, "Fig. 7shows the geometry of the proposed dual-sense CP antenna." Please add a space after 7.

Reviewer 2:

Dear authors,

In the manuscript, a co-aperture dual-sense circularly polarized antenna for simultaneous transmit and receive systems is proposed. I have some questions.

1. The manuscript should be well polished. There are several grammatical mistakes.

2. In Fig.6, it shoule be pointed that which curve represents axial ration and which curve represents s11. There are some curves in Fig. 13 and it also should be pointed what these curves represent.

3. I think the antenna is not suitable for simultaneous transmit and receive systems because the antenna's bandwidth is determined by a standard of 10-dB isolation. STAR antennas should have high isolations between the TX and RX ports. Please make a thorough review to check it.

4. Why the isolation is up to 39 dB at 2.46 GHz and the isolation are very low in other frequencies? What's the principle behind this?

Reviewers' comments:

Reviewer's Responses to Questions

**Comments to the Author**

1. Is the manuscript technically sound, and do the data support the conclusions?

Reviewer #1: Yes

Reviewer #2: Yes

2. Has the statistical analysis been performed appropriately and rigorously? 

Reviewer #1: Yes

Reviewer #2: Yes

3. Have the authors made all data underlying the findings in their manuscript fully available?

Reviewer #1: Yes

Reviewer #2: Yes

4. Is the manuscript presented in an intelligible fashion and written in standard English?

Reviewer #1: Yes

Reviewer #2: Yes

5. Review Comments to the Author

Reviewer #1: This paper proposes a compact design of dual-sense circularly polarized for simultaneous transmit (Tx) and receive (Rx) communication systems. The high Tx-Rx isolation can be achieved by tuning the feeding positions. There are some major issues that raise my concerns.

1. Undoubtedly, the isolation of the transmitting and receiving ports is a critical parameter for a STAR antenna. Although the article explains how to achieve high Tx-Rx isolation by merely adjusting the feed position, its decoupling mechanism is not explicitly described. To enhance the reader's comprehension, kindly provide a comprehensive description of this. Also, could you please state what challenges there are for the decoupling problem of dual circularly polarized antennas with the same aperture?

2. In contrast to the co-aperture dual-sense CP antennas that are comparable in size, the antenna proposed in the article possesses the most limited bandwidth. Could you please provide an explanation as to why?

3. More comprehensive results need to be provided so that readers can clearly understand the effectiveness of the proposed antenna, such as antenna efficiency, envelope correlation coefficient, and radiation pattern of both ports.

4. In Figure 12, it can be clearly observed that the tested S21 is better than the simulated one. Is this phenomenon coincidental? If not, then elucidate this.

5. There are many grammar mistakes in this paper. For example, The last paragraph of the introduction. "In this paper, the dual CP antenna with size miniaturization is concentrated. The proposed STR antenna employs 2 ×2 unit-cell MS as the co-aperture to achieve compact size. ". Page 4, third line of the first paragraph, "Fig. 7shows the geometry of the proposed dual-sense CP antenna." Please add a space after 7.

Reviewer #2: Dear authors,

In the manuscript, a co-aperture dual-sense circularly polarized antenna for simultaneous transmit and receive systems is proposed. I have some questions.

1. The manuscript should be well polished. There are several grammatical mistakes.

2. In Fig.6, it shoule be pointed that which curve represents axial ration and which curve represents s11. There are some curves in Fig. 13 and it also should be pointed what these curves represent.

3. I think the antenna is not suitable for simultaneous transmit and receive systems because the antenna's bandwidth is determined by a standard of 10-dB isolation. STAR antennas should have high isolations between the TX and RX ports. Please make a thorough review to check it.

4. Why the isolation is up to 39 dB at 2.46 GHz and the isolation are very low in other frequencies? What's the principle behind this?

6. PLOS authors have the option to publish the peer review history of their article (what does this mean?). If published, this will include your full peer review and any attached files.

Reviewer #1: No

Reviewer #2: No

---

## [Author Response · Author response to Decision Letter 0]

7 Apr 2024

Original Manuscript ID: PONE-D-24-05575

Original Article Title: “A compact co-aperture dual-sense circularly polarized antenna for simultaneous transmit and receive systems”

To: Reviewer

Re: Response to reviewer

Dear Reviewer,

We appreciate you for your precious time in reviewing our paper and providing valuable comments. It was your valuable and insightful comments that led to possible improvements in the current version. The authors have carefully considered the comments and tried our best to address every one of them.

We are uploading our point-by-point response to the comments, an updated manuscript with red highlighting indicating changes, and a manuscript without track changes.

Best regards,

 

Reviewer 1: This paper proposes a compact design of dual-sense circularly polarized for simultaneous transmit (Tx) and receive (Rx) communication systems. The high Tx-Rx isolation can be achieved by tuning the feeding positions. There are some major issues that raise my concerns.

Comment 1: Undoubtedly, the isolation of the transmitting and receiving ports is a critical parameter for a STAR antenna. Although the article explains how to achieve high Tx-Rx isolation by merely adjusting the feed position, its decoupling mechanism is not explicitly described. To enhance the reader's comprehension, kindly provide a comprehensive description of this. Also, could you please state what challenges there are for the decoupling problem of dual circularly polarized antennas with the same aperture.

Author response: The authors would like to thank the Reviewer for your very constructive comment.

Initially, the radiation principle of the proposed antenna is restated for Reviewer’s convenience. According to the discussion in Section “Single-port MS-based antenna”, the proposed antenna has two main parts including the CP source and the radiating element. Here, the radiating element is the MS layer, which determines the resonant frequency of the antenna. Meanwhile, as the MS layer is a combination of multiple symmetric unit cells, it cannot radiate CP wave itself. Thus, the CP source is necessary, and it is tuned accompanied by the MS to achieve good performance at the desired band.

Regarding the decoupling mechanism, the MS plays an important role in determining the inter-port isolation of the proposed design. Fig. 1R shows the current distributions on the CP source at 2.46 GHz for different cases of with and without MS.

Fig. 1R. Simulated current distributions at 2.46 GHz with and without the MS.

It can be seen obviously that with the presence of the MS, the field on the CP source is significantly redistributed and the intensity field is weak at the Port-2 feeding branch. This is due to the MS is designed to well resonate at 2.46 GHz. It can be verified by observing Fig. 4b in the revised manuscript, the modal significance is approximately 1 at 2.46 GHz. At this frequency, the electromagnetic field from the CP source tends to couple with the MS, rather than the Port-2 feeding branch, leading to high inter-port isolation. Out of the frequency band around 2.46 GHz, the field distribution on the CP source is changed significantly, as demonstrated in Fig. 2R.

Fig. 2R. Simulated current distributions on the patch at different frequencies.

As shown in Fig. 2R, the field distributions on the CP source are significantly different at different frequencies. At 2.4 and 2.5 GHz, strong currents are observed on the Port-2 feeding branch, resulting in high isolation (as seen in Fig. 8a of the revised manuscript). This is attributed to the MS, which doesn’t resonate at these frequencies due to the low modal significance values of less than 0.4 (shown in Fig. 4b of the revised manuscript). Thus, the isolation performance at these frequencies is worse than 2.46 GHz.

To conclude, the decoupling mechanism of the proposed MS-based antenna strongly depends on the MS. At the resonant frequency (determined by the MS), the feeding position of Port-2 is tuned to achieve the best isolation performance (shown in Fig. 9b of the revised manuscript).

With respect to the challenges for the decoupling of dual circularly polarized antennas with the same aperture, it is indeed much more difficult than decoupling dual CP antennas with different apertures (like two-port MIMO antenna). Here, the port spacing is small and the coupling is very strong due to the use of the same radiating aperture.

Author action: The decoupling mechanism is further discussed in Paragraph 2, Section “Dual-sense CP two-port MS-based antenna” of the revised manuscript. Figs. 1R and 2R are also included in the revised manuscript as Figs. 9 and 10. The challenging in decoupling of co-aperture antenna is mentioned in Paragraph 3, Section “Introduction”.

Comment 2: In contrast to the co-aperture dual-sense CP antennas that are comparable in size, the antenna proposed in the article possesses the most limited bandwidth. Could you please provide an explanation as to why?

Author response: In fact, the small antenna will have relatively low radiation resistance or high Q-factor, leading to very narrow BW. As this paper focuses on compact designs, there is a trade-off with operating BW. However, despite having the smallest overall dimensions, the proposed antenna can achieve high isolation and comparable gain as well. Note that for compact design, the decoupling is more difficult as the short distance between two ports. 

Author action: Further discussion about the performance of the proposed antenna is added to Paragraph 4, Section “Measurement results” of the revised manuscript.

Comment 3: More comprehensive results need to be provided so that readers can clearly understand the effectiveness of the proposed antenna, such as antenna efficiency, envelope correlation coefficient, and radiation pattern of both ports.

Author response: The authors would like to thank the Reviewer for your constructive comment.

Regarding the antenna efficiency, the simulated value is about 84%. In measurement, the antenna efficiency was not measured due to the limit of the chamber; however, it can still be estimated based on the realized gain measurements (Fig. 13). The gain difference between simulation and measurement is about 0.3 dBi. Thus, the calculated measured efficiency is about 76%.

Author action: The radiation efficiency is mentioned in Paragraph 2, Section “Measurement results”. The ECC is discussed in Paragraph 1, Section “Dual-sense CP two-port MS-based antenna”. Regarding the radiation pattern, as the antenna is symmetric, the radiation patterns with Port-2 excitation is like Port-1. Thus, the authors believe that showing radiation pattern with Port-1 is proper. This is highlighted in Paragraph 3, Section “Measurement results”.

Comment 4: In Figure 12, it can be clearly observed that the tested S21 is better than the simulated one. Is this phenomenon coincidental? If not, then elucidate this.

Author response: In fact, there always exists a difference between simulations and measurements. Sometimes simulated results are better and vice versa. Thus, the authors believe that this phenomenon is just coincidence. 

Comment 5: There are many grammar mistakes in this paper. For example, The last paragraph of the introduction. "In this paper, the dual CP antenna with size miniaturization is concentrated. The proposed STR antenna employs 2 ×2 unit-cell MS as the co-aperture to achieve compact size. ". Page 4, third line of the first paragraph, "Fig. 7shows the geometry of the proposed dual-sense CP antenna." Please add a space after 7.

Author response: The author would like to thank the Reviewer for pointing out our grammatical mistakes.

Author action: The grammar is thoroughly checked.

Reviewer 2: In the manuscript, a co-aperture dual-sense circularly polarized antenna for simultaneous transmit and receive systems is proposed. I have some questions.

Author response: The author would like to thank the Reviewer for pointing out our grammatical mistakes.

Author action: The grammar is thoroughly checked in the revised manuscript.

Comment 2: In Fig.6, it should be pointed that which curve represents axial ration and which curve represents s11. There are some curves in Fig. 13 and it also should be pointed what these curves represent.

Author response: The author would like to thank the Reviewer for the constructive comment.

Author action: Fig. 6 and Fig. 13 are modified in the revised manuscript.

Comment 3: I think the antenna is not suitable for simultaneous transmit and receive systems because the antenna's bandwidth is determined by a standard of 10-dB isolation. STAR antennas should have high isolations between the TX and RX ports. Please make a thorough review to check it.

Author response: The author would like to thank the Reviewer for the constructive comment. As reported in the literature, self-interference is the most critical challenge to the development of STAR systems and high isolation is needed, as considered in [1R]. However, the cancellation techniques should be deployed to all related domains, including antenna/propagation domain, analog domain and digital domain.

In this study, although the peak isolation is significantly high, at 39 dB, the 20-dB isolation bandwidth is about 10 MHz. Hence, the proposed design could be applied to STAR applications that need narrow-band operations, such as Zigbee with 2 MHz-channel and Bluetooth with 5 MHz-channel. Similar isolation performance is also observed in [20], which is designed for STAR communication. Thus, the authors believe that the proposed design could be suitable for STAR applications.

Author action: The possible application of the proposed antenna is added to Section “Conclusion”.

[1R] Chen, Y.; Ding, C.; Jia, Y.; Liu, Y. Antenna/Propagation Domain Self-Interference Cancellation (SIC) for In-Band Full-Duplex Wireless Communication Systems. Sensors 2022, 22, 1699.

Comment 4: Why the isolation is up to 39 dB at 2.46 GHz and the isolation are very low in other frequencies? What's the principle behind this?

Author response: The authors would like to thank the Reviewer for your very constructive comment. The decoupling mechanism of the proposed design is further investigated.

Initially, for Reviewer’s convenience, the radiation principle of the proposed antenna is restated. According to the discussion in Section “Single-port MS-based antenna”, the proposed antenna has two main parts including the CP source and the radiating element. Here, the radiating element is the MS layer, and the resonant frequency is determined by this layer. Meanwhile, as the MS layer is a combination of multiple symmetric unit cells, it cannot radiate CP wave. Thus, the CP source is necessary, and it is tuned accompanied by the MS to achieve good performance at the desired band.

Regarding the decoupling mechanism, the MS plays an important role in determining the inter-port isolation of the proposed design. Fig. 3R shows the current distribution on the CP source at 2.46 GHz with and without MS.

Fig. 3R. Simulated current distributions at 2.46 GHz with and without the MS.

It can be seen obviously that with the presence of the MS, the field on the patch is significantly redistributed and the intensity field is weak at the Port-2 feeding branch. This is due to the MS is designed to well resonate at 2.46 GHz. It can be verified by observing Fig. 4b in the revised manuscript, the modal significance is approximately 1 at 2.46 GHz. At this frequency, the electromagnetic field from the CP source tends to couple with the MS, rather than the Port-2 feeding branch, leading to high inter-port isolation. Out of the frequency band around 2.46 GHz, the field distribution on the CP source is changed significantly, as demonstrated in Fig. 4R.

Fig. 4R. Simulated current distributions on the patch at different frequencies.

As shown in Fig. 4R, the field distributions on the CP source are significantly different at different frequencies. At 2.4 and 2.5 GHz, strong currents are observed on the Port-2 feeding branch, resulting in high isolation (as seen in Fig. 8a of the revised manuscript). This is attributed to the MS, which doesn’t resonate at these frequencies due to the low modal significance values of less than 0.4 (shown in Fig. 4b of the revised manuscript). Thus, the isolation performance at these frequencies is worse than 2.46 GHz.

To conclude, the decoupling mechanism of the proposed MS-based antenna strongly depends on the MS. At the resonant frequency (determined by the MS), the feeding position of Port-2 is tuned to achieve the best isolation performance (shown in Fig. 9b of the revised manuscript).

Author action: The decoupling mechanism is further discussed in Paragraph 2, Section “Dual-sense CP two-port MS-based antenna” of the revised manuscript. Figs. 1R and 2R are also included in the revised manuscript as Figs. 9 and 10.

---

## [Decision Letter · Decision Letter 1]

13 May 2024

A compact co-aperture dual-sense circularly polarized antenna for simultaneous transmit and receive systems

PONE-D-24-05575R1

Dear Dr. Tran-Huy,

We’re pleased to inform you that your manuscript has been judged scientifically suitable for publication and will be formally accepted for publication once it meets all outstanding technical requirements.

Kind regards,

Maharana Pratap Singh, Ph.D.

Academic Editor

PLOS ONE

---

## [Editor Report · Acceptance letter]

25 Jun 2024

PONE-D-24-05575R1 

PLOS ONE

Dear Dr. Tran-Huy, 

I'm pleased to inform you that your manuscript has been deemed suitable for publication in PLOS ONE. Congratulations! Your manuscript is now being handed over to our production team.

Kind regards, 

on behalf of

Dr. Maharana Pratap Singh 

Academic Editor

PLOS ONE